# Functional Biomarkers for the Selenium Status in a Human Nutritional Intervention Study

**DOI:** 10.3390/nu12030676

**Published:** 2020-03-02

**Authors:** Sandra M. Müller, Christine Dawczynski, Johanna Wiest, Stefan Lorkowski, Anna P. Kipp, Tanja Schwerdtle

**Affiliations:** 1Department of Food Chemistry, Institute of Nutritional Science, University of Potsdam, 14558 Nuthetal, Germany; sandramueller@uni-potsdam.de; 2NutriAct-Competence Cluster Nutrition Research, 14467 Berlin-Potsdam, Germany; 3Institute of Nutritional Sciences, Friedrich Schiller University, 07743 Jena, Germany; christine.dawczynski@uni-jena.de (C.D.); johanna.wiest@web.de (J.W.); stefan.lorkowski@uni-jena.de (S.L.); anna.kipp@uni-jena.de (A.P.K.); 4Competence Cluster for Nutrition and Cardiovascular Health (nutriCARD), 07743 Halle-Jena-Leipzig, Germany; 5TraceAge – DFG research unit 2558, 07743 Potsdam-Berlin-Jena, Germany

**Keywords:** Se, selenoprotein P, GPx activity, cardiovascular disease, status markers

## Abstract

Soils in Germany are commonly low in selenium; consequently, a sufficient dietary supply is not always ensured. The extent of such provision adequacy is estimated by the optimal effect range of biomarkers, which often reflects the physiological requirement. Preceding epidemiological studies indicate that low selenium serum concentrations could be related to cardiovascular diseases. Inter alia, risk factors for cardiovascular diseases are physical inactivity, overweight, as well as disadvantageous eating habits. In order to assess whether these risk factors can be modulated, a cardio-protective diet comprising fixed menu plans combined with physical exercise was applied in the German MoKaRi (modulation of cardiovascular risk factors) intervention study. We analyzed serum samples of the MoKaRi cohort (51 participants) for total selenium, GPx activity, and selenoprotein P at different timepoints of the study (0, 10, 20, 40 weeks) to explore the suitability of these selenium-associated markers as indicators of selenium status. Overall, the time-dependent fluctuations in serum selenium concentration suggest a successful change in nutritional and lifestyle behavior. Compared to baseline, a pronounced increase in GPx activity and selenoprotein P was observed, while serum selenium decreased in participants with initially adequate serum selenium content. SELENOP concentration showed a moderate positive monotonic correlation (r = 0.467, *p* < 0.0001) to total Se concentration, while only a weak linear relationship was observed for GPx activity versus total Se concentration (r = 0.186, *p* = 0.021). Evidently, other factors apart from the available Se pool must have an impact on the GPx activity, leading to the conclusion that, without having identified these factors, GPx activity should not be used as a status marker for Se.

## 1. Introduction

Selenium (Se) is an essential trace element and is required for the synthesis of selenoproteins, such as glutathione peroxidases (GPx), thioredoxin reductases (TXNRD), iodothyronine deiodinases (DIO), as well as the transport protein selenoprotein P (SELENOP) [1,2,3]. For consumers, sources of Se include cereals, nuts and especially meat, but their Se content greatly differs depending on where they originate from [4,5,6]. The main Se species in foodstuffs are selenomethionine (SeMet) and selenocysteine (SeCys), both contributing to the metabolically available Se pool. The nutritional status describes not only the amount of a nutrient ingested by an individual, but also how it is retained and metabolized in the human body [4]. Thus, the ideal status marker for a particular nutrient responds to changes in the supply in a proportional manner [7]. 

In case of Se, the circulating Se content in plasma consists of the extracellular selenoproteins GPx3 and SELENOP [4], Se-containing proteins with unspecifically incorporated SeMet instead of Met, and different Se species bound to glutathione or albumin. The plasma level of Se is pre-eminently dependent on the baseline supply: in cases where provision of Se is low (substantially below 55 µg/d according to US RDA [8]), the plasma level rises with any Se species added to the available Se pool [9,10]. In the case of already adequately supplied individuals, it has been shown that plasma Se levels are dependent on the consumed Se species. While inorganic Se species increase plasma Se levels beneath notice, SeMet elevates the level of plasma Se significantly [11,12]. The reason for this is the enhanced elimination of inorganic species, whereas SeMet is retained by unspecific incorporation into proteins. Therefore, it provides a non-regulated Se storage in the body. As the SELENOP and GPx3 plasma concentrations correlate inversely with the severity of Se deficiency [13], they can be used as nutritional biomarkers for Se in an adequately wide concentration range until they are fully optimized and reach a plateau. SELENOP is a glycoprotein with up to 10 SeCys residues which acts as the primary transporter of Se in plasma, accounting for 60% of plasma Se [14,15,16]. Hence, SELENOP is linearly related to plasma Se [17,18], yet it only reflects Se intake until the need for selenoprotein synthesis is met [12,14]. Se concentrations around 80–90 µg/L plasma or serum are considered to be necessary for maximal selenoprotein expression [4,10]. While the cellular components of blood mirror Se intake over a longer time period, plasma or serum are easier to handle and mainly reflect short-term changes in Se status [19]. In Germany, plasma Se levels of adults range between 52 and 115 µg/L [20,21] indicating that individuals in the lowest tertile are not meeting the requirement. 

In contrast to European countries, Keshan disease and Kaschin-Beck disease, which are both associated with an unusually low Se supply of below 10 µg per day, are only observed in regions of China and in parts of East Africa [22,23,24]. Keshan disease is characterized by a cardiomyopathy which indicates that the heart belongs = tissues which are susceptible towards a limited Se supply. A similar phenotype can be observed in patients with rare mutations in TXNRD2 (p.Ala59Thr and p.Gly375Arg) who suffer from dilated cardiomyopathy [25]. Furthermore, low plasma Se and SELENOP concentrations are associated with the risk of developing cancer [26,27] and cardiovascular diseases [28]. The link between the Se status and cardiovascular disease particularly warrants further investigation [29,30]. 

Therefore, in this study we investigated serum Se, GPx, and SELENOP levels of participants in the MoKaRi (modulation of cardiovascular risk factors) cohort, a study originally designed to explore how a nutrition intervention combined with physical activity affects cardiovascular risk markers. The study participants already had an increased risk of developing cardiovascular disease at baseline. The great advantage of the study is the frequent sampling of blood during a well-controlled change in nutritional habits and physical activity towards a healthier lifestyle. This design allows for intra-individual comparison during the study period. As short-term changes in Se status should be monitored, serum samples were chosen instead of whole-blood. Both arms of the cohort underwent the aforementioned change in lifestyle, while one group was additionally supplemented with fish oil (henceforth referred to as + fish oil group) and the other was not (- fish oil group). However, as fish oil does not provide any Se, we did not expect any differences between those two groups. We aimed to address the question of how sensitive the Se status is to changes in nutritional behavior in a population with Se intake levels in the adequate to suboptimal range. Besides this, we wanted to see how these changes affect the different components of the available set of markers for Se status (serum Se levels, SELENOP, GPx activity) and whether these markers are equally suitable and well-correlated. As Se appears to be of relevance in the context of cardiovascular disease, characterizing the Se status of the study participants who are at risk of developing cardiovascular disease might provide new ideas about preventive strategies in the future.

## 2. Materials and Methods 

### 2.1. Participants

Participants of both sexes aged between 32 to 76 years with plasma low-density lipoprotein (LDL) cholesterol concentrations ≥ 3 mmol/L were enrolled after written informed consent. The MoKaRi study was completed by 51 subjects (dropout: nine participants). The study collective was randomly separated into two groups, with 30 participants in each group. The fish oil group was finished by six males and 19 females, age between 33 and 76 years (58 ± 13 years) and the group without fish oil comprised nine males and 17 females, age between 32 and 76 years (61 ± 11 years). Randomization was conducted with R statistics (package blockrand, block size 8).

The exclusion criteria comprised intake of lipid-lowering medications and glucocorticoids, gastrointestinal diseases, known allergies or food intolerances, known familial hypercholesterolemia, intake of additional dietary supplements (e.g., fish oil capsules or vitamin E), body mass index (BMI) ≥ 25 kg/m^2^, pregnancy, lactation, regular abuse of alcohol or drugs, and patient’s request, or if patient compliance with the study protocol was doubtful. 

During the course of the MoKaRi trial, any sporadic and systemic use of medications in the context of other diseases was allowed if it did not interfere with the study course. Participants had to document intake and dosage of all medications. The systemic intake of medications in the context of other diseases should not be changed over the study period.

### 2.2. Study Design and Diet

The MoKaRi concept was designed to improve nutritional behavior by means of the following three approaches: (i) personal menu plans for implementing a “cardioprotective diet”, (ii) regular, individual nutritional counselling, and (i) incentives, such as (a) information about healthy nutrients, (b) the possibility of participating in cardio-protective circle training, (c) provision of study foods, e.g., native olive oil, nuts mixture, (d) regular feedback on the course of study parameters, and (e) activities to encourage group feeling. 

The MoKaRi trial is designed as a randomized, single-center intervention study in parallel design with two arms (+ fish oil group: prepared menu plans with 3 g long-chain *n*-3 fatty acids by fish oil per day; *n* = 26 vs. - fish oil group: prepared menu plans without fish oil; *n* = 25). The dietary regimes of both groups are isocaloric.

The MoKaRi concept is based on prepared menu plans, which ensured: 1Adequate intake of energy, carbohydrates, protein and fat according to the guidelines of the German Society of Nutrition [31];2Desirable intake of saturated fatty acids (SFA, ≤ 7 % of daily energy), monounsaturated fatty acids (MUFA, ≥ 10 % of daily energy), polyunsaturated fatty acids (PUFA, ≥ 10 % of daily energy) and long-chain n-3 PUFA (≥ 500 mg/d);3Encouraged consumption of vegetables, fruits and cereals;4Intake of > 40 g/d dietary fiber,;5Reduced intake of salt and sugar reduction;6Reduced intake of (highly) processed, calorie-rich, nutrient-poor foods (e.g., fast food, convenience products);7Optimized intake of vitamins, minerals and trace elements by commercially available foods; and8Encouraged physical activity.

The menu plans were adapted to participant’s individual energy requirements as dictated by age, gender and level of physical activity. Se intake of the menu plans is unknown and cannot be estimated since Se values are only poorly recorded in databases.

The MoKaRi study started with a run-in period to assess and document the dietary habits of the study participants using a food frequency protocol (FFP) over seven days. 

In the further course of the MoKaRi trial, fasting blood sampling, assessment of health status, inclusive visualization of the cardiovascular risk profile, supply of 14 new menu plans embedded in personalized nutritional counselling units and provision of study foods were conducted every 14 days during the 20 weeks of the trial (i.e., 11 times). The follow-up period of 20 weeks was split into two periods (every 10 weeks) for fasting blood sampling. The participants were expected to maintain their diet but did not get further nutritional counselling or new menu plans.

The study protocol has been approved by the responsible ethical committee of the Friedrich Schiller University Jena (number 4656-01/16) and the MoKaRi study was registered before launch at ClinicalTrials.gov (identifier NCT02637778; https://clinicaltrials.gov/ct2/show/NCT02637778).

### 2.3. Laboratory Measurements 

#### 2.3.1. Total Selenium

Before preparation, serum samples were stored at -80 °C. Samples were prepared as published [32]. Briefly, samples were thawed within 1.5 h and diluted (1:5) with a mixture of 1-butanol (5%), sodium EDTA (0.05%), Triton-X 100 (0.05%), and NH_4_OH (0.25%) in deionized water. Rh and ^77^Se were added as internal standards. Total selenium was determined by isotope dilution analysis *via* ICP-MS/MS (Agilent ICP-QQQ 8800, Waldbronn, Germany). Reference sera (ClinChek serum Level 1 and 2 by RECIPE Chemicals + Instruments, Munich, Germany; Seronorm by Sero AS, Billingstad Norway) were analyzed with every measurement. 

#### 2.3.2. GPx Activity

As described before [33], a NADPH-consuming glutathione reductase-coupled assay was applied for assessment of GPx activity (glutathione reductase obtained from Roche Diagnostics, Basel, Switzerland), while hydrogen peroxide (Merck, Darmstadt, Germany) was used as substrate. Here, 17.5 µL of diluted serum (1:5) was used and absorbance was measured as triplicate using a microplate reader (Synergy H1, BioTek, Bad Friedrichshall, Germany). GPx activity is expressed as mU/mg protein.

#### 2.3.3. SELENOP

For the quantification of SELENOP, an enzyme-linked immunosorbent assay (ELISA) was handled according to the manufacturer’s instructions (selenoOtest ELISA, selenOmed, Berlin, Germany).

### 2.4. Statistical Analysis

The statistical analysis was performed by Prism GraphPad 8 and R Studio version 1.1.463. There was no difference between males and females for total Se, GPx and SELENOP as tested with Mann-Whitney U-test, thus data of both genders were combined for higher power in further evaluations. Also, the comparison between control and intervention was tested by Mann–Whitney U-test. For the comparison of time-dependent changes, Skilling’s Mack test was performed with Dunn’s post-hoc test. Statistical significance is referred to *p*-values smaller than 0.05. 

## 3. Results 

Blood serum of 51 participants of the MoKaRi cohort was analyzed for total selenium (Figure 1). At the beginning of the study, mean of total Se for the - fish oil group was 83.0 ± 19.3 µg/L, with a minimum value at 53.5 µg/L and a maximum value of 138.4 µg/L. The mean of total Se for the + fish oil group was 78.7 ± 12.0 µg/L, with a range from 63.2 to 120.3 µg/L. During the intervention time of 20 weeks, the serum selenium concentrations ranged between 67.0 and 94.7 µg/L for the - fish oil group and between 66.6 and 95.6 µg/L for the + fish oil group. At the end of the intervention, the participants had mean selenium levels of 77.7 ± 10.4 µg/L in the - fish oil group (range: 62.0–106.8 µg/L), while the + fish oil group showed levels of 74.4 ± 7.8 µg/L (range: 57.6–87.6 µg/L). After ten weeks of follow-up (week 30), total serum selenium increased to 95.4 ± 15.8 µg/L for the – fish oil and to 95.3 ± 11.8 µg/L for the + fish oil group. The values dropped for both groups at the end of follow up (week 40): the - fish oil group was at 69.7 ± 12.3 µg/L, whereas the individuals of the + fish oil group were at 71.9 ± 10.7 µg/L. 

In further experiments, SELENOP concentrations as well as GPx activities have been determined for selected time points, i.e., the beginning of the intervention (0 weeks), at half time (10 weeks), the end of the intervention (20 weeks), and the last follow-up visit (40 weeks). Table 1 summarizes the results of SELENOP and GPx activity measurements in comparison with total Se for the mentioned timepoints, while Figure 2 visualizes these data. For total Se (Figure 2a), SELENOP (Figure 2b), and GPx activity (Figure 2c), no significant differences were found between the - fish oil group and the + fish oil group. However, statistically significant time-dependent changes between the visits were observed. Mean SELENOP increased during the intervention period in both groups, reaching its highest level at the end of the intervention with 5.9 ± 0.9 mg/L for the – fish oil group (week 0 vs. week 20: *p* = 0.0008) and 5.7 ± 1.2 mg/L for the + fish oil group (week 0 vs. week 20: *p* = 0.0021). The follow-up visit after 40 weeks revealed a slight decline in mean SELENOP to 5.7 ± 1.2 mg/L for the - fish oil group and 5.5 ± 1.3 mg/L for individuals supplemented with fish oil. The behavior of increases and decreases for mean SELENOP was not consistent for all individuals, as can be taken from the minimum and maximum values (Table 1, ranges).

Levels of GPx activity started with 384.0 ± 82.5 U/L for the - fish oil group and 425.9 ± 150.6 U/L for the + fish oil group. For both groups, mean GPx activity reached a peak at the half-time visit (week 0 vs. week 10: *p* < 0.0001). After 40 weeks, the - fish oil group had a mean GPx activity of 521.2 ± 117.8 U/L (week 0 vs. week 40: *p* = 0.0002), while the + fish oil group was at 520.7 ± 139.0 U/L (week 0 vs. week 40: *p* = 0.0061).

When plotting SELENOP concentration against total Se concentration, a moderate positive monotonic correlation between these two biomarkers was found (Figure 3a; *p* < 0.0001). Interestingly, only a weak linear relationship can be observed for GPx activity versus total Se concentration (Figure 3b), with r = 0.186 (*p* = 0.021). A more distinct correlation (r = 0.338; *p* < 0.0001) was found for GPx activity vs. SELENOP concentration (Figure 3c). 

When dissecting the cohort into two groups based on the initial serum Se content (< 80 µg/L and > 80 µg/L), a clear pattern can be identified for differences in serum Se content compared to baseline values (Figure 4a). A serum Se content of 80 µg/L was chosen as a cut-off because this is the concentration presently defined to optimize serum selenoprotein concentrations [10]. In the course of the study, the serum Se concentrations showed a trend for a decrease in the group with high initial serum Se (mean: 92.9 ± 14.4 µg/L; week 10 vs. week 40: *p* = 0.0854). In the group with low initial serum Se (mean: 71.7 ± 7.5 µg/L) a decline was only identified in the follow-up (40 weeks; week 10 vs. week 40: *p* = 0.0003). Consequently, the number of individuals with < 80 µg/L serum Se increased during the study. After 10 weeks of intervention, six participants dropped below 80 µg/L Se. At the end of the intervention (20 weeks), nine individuals and, for the follow-up, eventually 11 individuals had serum Se concentrations of lower than 80 µg/L. For the group with initially low serum Se concentrations, four participants showed > 80 µg/L serum Se after 10 weeks and, after 20 weeks, there were six individuals with more than 80 µg/L serum Se. This effect was reversible as, in the follow-up, the serum Se concentrations fell back again to < 80 µg/L. In contrast to serum Se values, there were no differences between SELENOP concentrations (Figure 4b) or serum GPx activity (Figure 4c) in comparison to their baseline values or between the groups with baseline Se serum values below or above 80 µg/L. 

## 4. Discussion

In this study, the control group and the intervention group differed from each other only in the supplementation with fish oil. Fish oil is neither enriched with selenium species nor is it a noteworthy selenium source, consequently, differences between both study groups were not expected and were not found. 

Our data show that GPx activity does not correlate with total Se (Spearman r = 0.02, Figure A1b) for individuals with > 80 µg/L serum Se, which is in line with the literature [10]. However, we observed a decrease in Se concentration for participants who are initially adequately supplied with Se (> 80 µg/L in serum) during the intervention period of 20 weeks. Concurrently, the GPx activity increases in both groups (- fish oil, + fish oil) regardless of the initial serum Se concentration, whereas the SELENOP content did not change substantially until week 20 as compared to baseline. This is surprising, as SELENOP is expected to be the more convincing biomarker in this concentration range. It has been supposed that a serum Se concentration > 80–90 µg/L will lead to a plateau of GPx3 activity [10], thus it is unexpected to see an increase in GPx activity beyond that “threshold”. Due to the limited relationship between GPx activity and serum Se, it seems plausible that other factors might have influenced GPx activity independently from serum Se concentration. 

In human serum, the present isoform of GPx is GPx3. For GPx1, the GPx1-198Leu genotype has been associated with reduced enzyme activity [34], but could not be confirmed by other studies [35,36]. As the measured Se-dependent enzyme activity of GPx in serum reflects total selenium up to a concentration of 80 µg/L poorly (Spearman r = 0.26, Figure A1a), it is possible that this observation is due to genetic variants in GPx3. Hence, it is conceivable that certain GPx3 variants have a higher or lower enzyme activity, independent from the selenium level. Indeed, various genotypes of GPx3 are known [37]. Still, not only the genetic variation, but also allosteric effects, must be considered when evaluating GPx activity [38]. The biosynthesis of GPx3 in the kidneys depends on the megalin-mediated endocytosis of SELENOP into the renal proximal tubule epithelial cells [39]. Here, genetic variants play an important role, as plasma Se concentrations depend on single-nucleotide polymorphisms (SNP) of SELENOP, i.e., Ala234Thr and r25191 g/a. In a previous study by Méplan et al., it has been observed that Se supplementation led to a significant increase in GPx activity, according to which genotype was present for r25191 g/a. Furthermore, the genotype decided upon an interaction of GPx3 activity with gender, resulting in lower GPx activity for AA homozygous males [40]. 

In addition, it has to be kept in mind that all study participants started with plasma LDL cholesterol concentrations ≥ 3 mmol/L, which is above optimal values (ranging below 2.6 mmol/L). At the end of the intervention, a significant reduction in BMI was found for the study participants (-fish oil: Δ -1.5 ± 1.1 kg/m², + fish oil: Δ -1.0 ± 0.6 kg/m²). Based on gene expression experiments, it has been hypothesized that GPx3 is a direct estrogen receptor α target gene in white adipose tissue [41]. Moreover, an association between obesity and reduced expression of GPx3 has been demonstrated [42]. Accordingly, the increase in GPx activity could be the result of the decline in adipose tissue. In the context of obesity being regularly associated with chronic inflammation [43], the enhanced level of GPx activity and SELENOP content could contribute to the prediction of inflammation, as a study in young adults recently demonstrated the inverse relationship between selenoproteins and hepcidin, a protein of iron homeostasis and an inflammation marker [44]. It also remains unclear why the SELENOP concentration rises during the study, while serum Se drops. One possibility is that this observation reflects the response to the decreasing serum Se concentration in order to maintain the supply of extra-hepatic tissues with Se [45]. It has been demonstrated that SELENOP has endothelium-protecting activity in subjects with sepsis-related endothelium-dysfunction by binding to the endothelium, therefore depleting the plasma of SELENOP [46]. With respect to the duration of the intervention study and the rapid turnover of SELENOP [14,47], it is unlikely that the elevated SELENOP level is related to a release of the protein from the endothelium in our case, however, this possibility cannot be excluded either, especially when considering the fact that our study participants were chosen for their elevated risk of cardiovascular diseases.

The limitations of the study include a misbalance in gender (a lower number of male participants) and the wide range of ages, which makes stratification challenging and therefore conceals presumably existing confounders. Although the participants followed strict menu plans, the Se intake is unknown and cannot be estimated because Se values are only poorly recorded in databases. These hindrances aggravate the evaluation of the observed effects.

## 5. Conclusions

The Se intake levels of our study population is considered as adequate to suboptimal. Ideally, a status marker for a particular nutrient such as Se responds to changes in the supplementation in a proportional manner and reflects the coverage of any physiological requirements (= effect biomarker) [48]. In our study, markers for the Se status comprise serum Se concentrations, SELENOP levels, and the GPx activity. We report that the linear relationship between serum Se and SELENOP is not as strong as expected for participants with initially low serum Se. In addition, the sensitivity of SELENOP as a marker for Se intake (reflected as serum Se concentration) is rather low, as serum Se levels increase earlier than SELENOP levels. Furthermore, the cause of the observed discrepancy between the serum Se level and the GPx activity remains a matter of speculation. Clearly, other factors apart from the available Se pool must have an impact on the GPx activity, leading to the conclusion that such factors must not only be identified, but the respective results have to be adjusted for these as of yet unknown modifiers in advance of using GPx activity as a status marker for Se. Our findings support the need for further analyses regarding, e.g., markers of early inflammation processes and blood lipids, in order to approach an explanation. 

## Figures and Tables

**Figure 1 nutrients-12-00676-f001:**
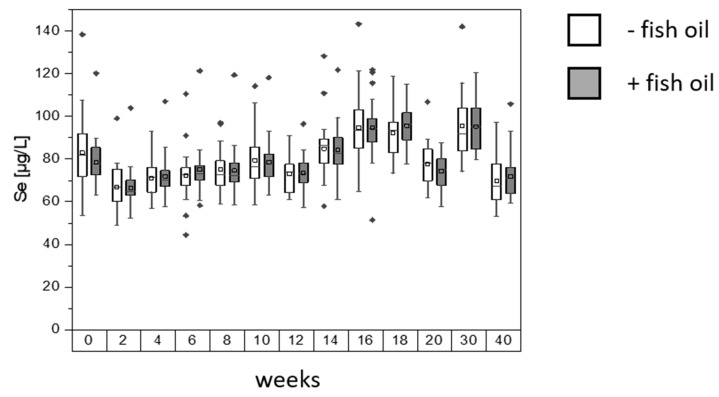
Serum selenium concentration of the study participants. Box plots are shown with ± 1.5 x interquartile range; square: mean; line: median. Group sizes differed between sampling visits: - fish oil group size between 20 and 25; + fish oil group size between 21 and 26 for each visit. Start of intervention in week 0, end of intervention in week 20, first follow up (after 10 weeks) in week 30, second follow up (after 20 weeks) in week 40.

**Figure 2 nutrients-12-00676-f002:**
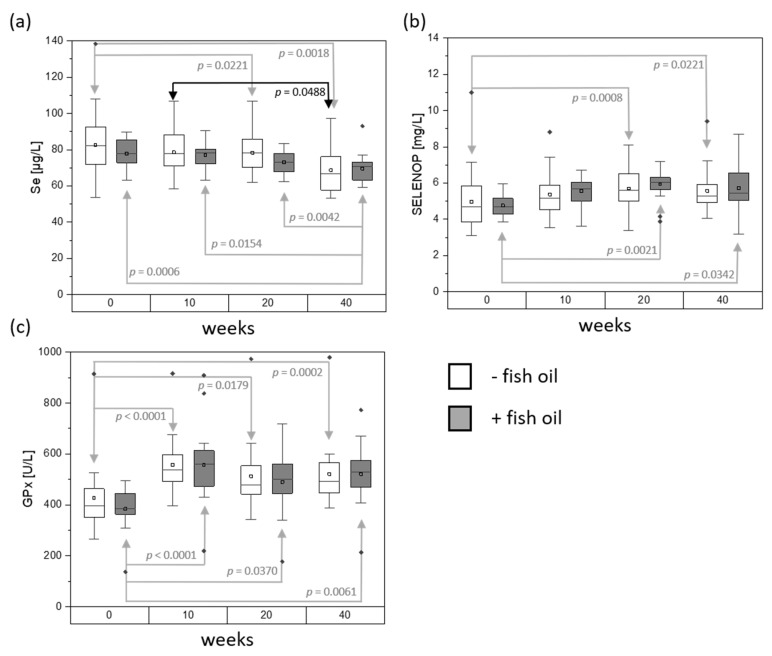
(**a**) Total selenium concentration, (**b**) SELENOP concentration and (**c**) GPx activity in the serum of the - fish oil group and + fish oil group after 0, 10, 20, and 40 weeks. - fish oil group *n* = 18, + fish oil group *n* = 19. Shown are box plots with ± 1.5 x interquartile range; square: mean; line: median. Friedman test followed by Dunn’s multiple comparisons.

**Figure 3 nutrients-12-00676-f003:**
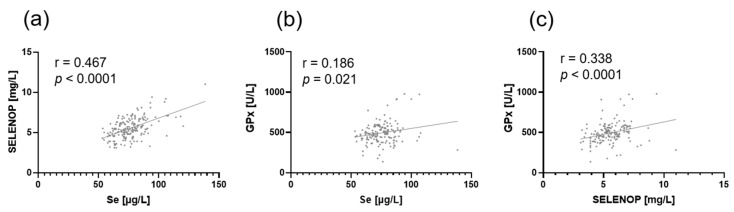
Correlation between (**a**) SELENOP and total Se concentration, (**b**) GPx activity and total Se concentration, and (**c**) GPx activity and SELENOP concentration. Spearman‘s procedure was used to calculate the correlation coefficient.

**Figure 4 nutrients-12-00676-f004:**
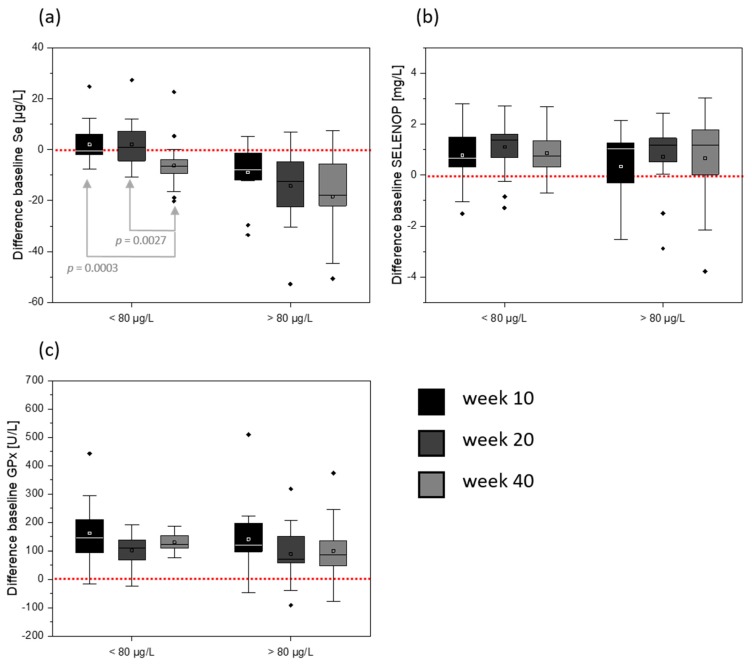
Difference to baseline Se concentration (**a**), baseline SELENOP concentration (**b**), and baseline GPx activity (**c**) regarding the initial serum Se concentration. < 80 µg/L: *n* = 22, > 80 µg/L: *n* = 15. Shown are box plots with ± 1.5 x interquartile range; square: mean; line: median; red line: baseline; dots indicate outliers.

**Table 1 nutrients-12-00676-t001:** Age, total selenium, SELENOP concentration and GPx activity in the - fish oil and + fish oil group at weeks 0, 10, 20, and 40. - fish oil group *n* = 18, + fish oil group *n* = 19. *p*-value of difference between groups, as determined by Mann–Whitney U-test.

		Age [years]	Se [µg/L]				SELENOP [mg/L]				GPx Activity [U/L]			
	Weeks		0	10	20	40	0	10	20	40	0	10	20	40
- fish oil		61	77.8	77.1	73.1	69.6	4.8	5.6	5.9	5.7	384.0	557.0	490.2	521.2
	SD	11.3	7.1	7.4	6.1	8.2	0.6	0.8	0.9	1.2	82.5	152.1	114.7	117.8
	Range	32–76	63.3–89.7	63.3–90.7	62.4–83.4	59.4–93.0	3.9–5.9	3.6–6.7	3.09–7.2	3.2–8.7	136.8–495.5	219.4–909.1	177.3–716.8	213.5–772.8
	Weeks		0	10	20	40	0	10	20	40	0	10	20	40
+ fish oil		57	83.3	79.2	78.6	67.8	5.1	5.5	5.7	5.5	425.9	557.5	512.8	520.7
	SD	13.4	21.1	13.9	11.1	11.9	2.0	1.5	1.2	1.3	150.6	123.0	146.1	139.0
	Range	33–76	53.5–138.5	58.6–106.6	62.0–106.8	53.2–94.4	3.1–11.0	3.5–8.8	3.4–8.1	4.1–9.4	267.0–914.5	396.4–916.8	342.2–973.1	388.2–979.5
	*p*-Value		0.4750	0.9339	0.1131	0.3090	0.7575	0.3014	0.3618	0.3224	0.7206	0.9052	0.7722	0.4028

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
