# Peer review of "Functional Biomarkers for the Selenium Status in a Human Nutritional Intervention Study"

_nutrients, 2020, doi:10.3390/nu12030676_

Round 1

Reviewer 1 Report

The manuscript "Functional biomarkers for the selenium status in the MoKaRi cohort" demonstrates the results of an interestingly planned experiment aimed at testing how the intervention in nutrition, combined with physical activity, affects cardiovascular risk markers. In the study, the authors also refer to the suitability of selenium-associated markers (serum Se levels, P-selenoprotein and GPx activity) as indicators of selenium status.

I find this manuscript valuable and interesting, but nevertheless I have a few minor comments about it.

The abstract should be supplemented with one or two main conclusions.

Keywords should differ from the words used in the title, so I suggest giving up the word "biomarkers", and to replace "selenium" with the symbol of this trace element "Se".

In the “Participants” section of the “Materials and Methods” chapter, apart from the age range of the study participants, also their total number and gender should be added. Additionally, please supplement the “Study Design and Diet” subsection with information whether in the MoKaRi studies both experimental groups (zero sample group and fish oil group) were similarly differentiated in terms of gender, or whether one of them was clearly dominated with women or men.

In the “Results” chapter, when discussing Fig.4, at least one sentence should refer to Fig.4 b and 4c.

Author Response

Thanks so much for the valuable comments, please see the enclosed file .

Reviewer 2 Report

The review concerns the manuscript entitled "Functional biomarkers for the selenium status in the MoKaRi cohort".

The aime of the study was to assess the correlations between Se concentrations (at various time points) and the levels of several selenoproteins.

This is an interesting study that focuses on an important topic. The study goal has been achieved. The results are presented in a systematic and understandable way. The conclusions are logical.

I have comments on Introduction.

Although I understand the authors' interest in a particular problem, they should ephasize the clinical implications of their study more. The authors only mention Keshan diseases, and yet they analyze data from a cohort, which was collected for the assessment of cardiovascular risk.

First, it would be good to show more separate paragraphs in the introduction. This section seems a bit chaotic.

The first sentence is OK. Then it should be written that it is not clearly established what are the correlations between Se levels and selenoproteins. And this is important because Se plays important roles for human health and is known as a strong antioxidant. Understanding the mechanisms of cerrelations of Se and selenoproteins would understand the pathogenesis of many diseases assiciated with increased oxidative stress and low or high levels of Se ("U" effect). An example could be ...... (Here you can also mention Keshan's diseases).

I suggest looking at publications, e.g.: Ann Biol Clin (Paris) 2016, 1126; Nutrients 2019, 11, 1028;  Br J Nutr 2015, 113, 249-258;  Nutrients 2019, 11, 2298.

Author Response

Thanks so much for the valuable comments. Please see our response in the attached file. Kind regards

Reviewer 3 Report

General comments

An interesting study comparing biomarkers of Se status. The dataset is complete and appears to be examined from all angles; the discussion is well thought out.

Selenium can be measured in whole blood, serum, or plasma. Whole blood represents Se status over a longer time period, while serum and plasma are typically representative over the short-term. Perhaps the lack of a strong correlation among GPx, SELENOP, and serum Se may be related to the sample type in which they were measured? It might be interesting to include some discussion of results found in other studies using different sample types (i.e. whole blood).

There are no P-values in the Results text anywhere, which makes it difficult to interpret if the change being described was significant or not. P-values need to be provided when claims are being made about changes between treatments or over time.

Comments by line

Suggestions for additions to the text are noted in bold

Line 14: …sufficient dietary supply is not…

Line 20: …comprising of fixed menu…

Line 21: replace ‘blood’ with ‘serum’

Line 48: Glutathione peroxidase is abbreviated here as GPx3. However, throughout the rest of the text, GPx is used as an abbreviation for glutathione peroxidase. I encourage being consistent with abbreviations to avoid confusion among GPx forms.

Line 54: Please specify in what fluid (serum, plasma, whole blood, etc) Se concentrations of 80-90 ug/L are adequate.

Line 66: I suggest mentioning here that plasma/serum only reflects short-term Se status.

Line 68: This is the first mention of the MoKaRi cohort in the text body. Please define the abbreviation here. Also, if there is an original reference for this study elsewhere, it might be useful to include it here.

Line 74-78: It is mentioned later in the paper that no differences were expected due to fish oil treatment; it would be nice to have that stated here as well. Hypotheses for the other objectives would be useful here as well.

Materials and Methods section 2.1: Please include here how many participants were enrolled in the study

Line 99: The study was randomized, but no mention of any blocking factors. Were the treatments blocked by gender? Please indicate how many males and females were in each treatment.

Line 105: It appears the citation for the German Society of Nutrition is missing

Line 124: It is unclear exactly what the follow-up period was. Please explain if the participants were expected to maintain their diet on their own or go back to previous dietary habits.

Line 139, 141: change “has been” to “was”. The use of “has been” seems like that is not what was done for the current study.

Line 153: Statistical significance indications (e.g. for *, **, etc) were defined here, but do not appear in the text, tables, or figures anywhere.

Line 162-165: The sentence refers to a one-week follow-up, which is assumed to be after the 20-week treatment period, but this time-point is referred to as week 30, which would mean this is a 10-week follow-up. Same with the two-week follow-up, which is referred to as week 40. A similar confusion is noted on line 171-172 for Figure 1 description. From the materials and methods, it appears there should be 10-week and 20-week follow-ups. Please check these numbers and change/clarify as necessary.

Line 208: Please explain why 80 ug/L was chosen as a cut-off value for dividing the data set.

Line 209 – 212: It is stated that the serum Se decreases more dramatically in the group with high initial Se, but the Figure does not denote a significant difference among the three time-points for the high Se group, while there is a significant difference (P < 0.05) for the group with low initial serum Se. I don’t think a claim about a more dramatic decrease can be made without significant changes. P-values should be added to the text for clarity.

Line 216: the semi-colon following “20 weeks;” should be a comma.

Line 230-231: Do the authors have an idea of why serum Se decreased during the dietary intervention? Could the treatment diets have been lower in Se than the diets consumed prior to the study? Do the treatment diets meet the Se requirement?

Line 233-234: Please add a reference supporting the idea that SELENOP is a convincing biomarker

Line 244: the word “independently” should be “independent”

Line 275: change the word “since” to “because.” “Since” refers to a passage of time, and “because” refers to a cause and effect.

Line 274-276: It would have been useful to know Se intakes were not measured when reading the materials and methods. I think this statement could be moved to that section or at least earlier in the Results/Discussion sections.

Line 282 – 283: Because the authors are making a conclusion about correlations between serum Se and SELENOP for the divided dataset, this relationship needs to be mentioned in the results section. Only the overall correlation for the entire dataset is presented in the body of the paper (3a).

Line 291: The appearance of blood lipids in the last sentence of the conclusion was surprising. I could not find any discussion of why measuring blood lipids would be useful for helping to explain the current data.

Author Response

Many thanks for the very valuable comments and suggestions. Please find our Response enclosed in the attachment.
